# Microneedle-Assisted Transdermal Delivery of Lurasidone Nanoparticles

**DOI:** 10.3390/pharmaceutics16030308

**Published:** 2024-02-22

**Authors:** Ariana Radmard, Ajay K. Banga

**Affiliations:** Center for Drug Delivery Research, Department of Pharmaceutical Sciences, College of Pharmacy, Mercer University, Atlanta, GA 30341, USA; ariana.radmard@live.mercer.edu

**Keywords:** transdermal delivery, microneedle patches, PLGA nanoparticles, controlled drug release, chemical enhancer, in vitro permeation testing

## Abstract

Lurasidone, an antipsychotic medication for schizophrenia, is administered daily via oral intake. Adherence is a critical challenge, given that many schizophrenia patients deny their condition, thus making alternative delivery methods desirable. This study aimed to deliver lurasidone by the transdermal route and provide therapeutic effects for three days. Passive diffusion was found to be insufficient for lurasidone delivery. The addition of chemical enhancers increased permeation, but it was still insufficient to reach the designed target dose from a patch, so a microneedle patch array was fabricated by using biodegradable polymers. For prolonged and effective delivery, the drug was encapsulated in Poly (lactic-co-glycolic acid) (PLGA) nanoparticles which were made using the solvent evaporation method and incorporated in microneedles. Effervescent technology was also employed in the preparation of the microneedle patch to facilitate the separation of the needle tip from the patch. Once separated, only the needle tip remains embedded in the skin, thus preventing premature removal by the patient. The microneedles demonstrated robust preformation in a characterization test evaluating their insertion capacity, mechanical strength, and the uniformity of microneedle arrays, and were able to deliver a dose equivalent to 20 mg oral administration. Therefore, the potential of a transdermal delivery system for lurasidone using microneedles with nanoparticles was demonstrated.

## 1. Introduction

Schizophrenia is a complex mental disorder characterized by psychosis, hallucinations, changes in consciousness and perception, a lack of interest, and cognitive deficits. It globally impacts approximately 24 million individuals worldwide [1]. Collectively, these symptoms result in impaired functioning in various aspects of life, including work, school, and social interactions [1,2]. It ranks among the top ten causes of long-term disability and necessitates extensive mental health care. Individuals with schizophrenia often struggle to recognize their illness or the necessity for treatment, and this lack of insight is linked to various adverse outcomes. These include nonadherence to treatment, a heightened risk of relapse, more severe psychological symptoms, and an overall poor prognosis for the individual’s condition [3]. Antipsychotic medications have notable limitations. Initially, their effectiveness is observed in approximately 50% of patients. Additionally, they are associated with significant metabolic side effects and can potentially result in issues like sexual dysfunction or agranulocytosis [4]. One class of drugs to treat schizophrenia is atypical antipsychotics, which inhibit the HT2A serotonin receptor and dopamine D2 receptor. Lurasidone, classified as a novel atypical (second-generation) antipsychotic medication, has gained approval for the treatment of adult schizophrenia. Lurasidone is superior to other atypical antipsychotics as it exhibits the highest affinity for the 5-HT7 receptor, along with offering enhanced therapeutic advantages to treat schizophrenia. Further, it has a high safety profile due to the absence of cardiovascular side effects and limited adverse effects on weight or metabolic profiles [5]. Lurasidone, when administered orally, undergoes extensive metabolism. As a result, only a substantial portion of the administered dose becomes available in the systemic circulation [6]. Moreover, the cytochrome P450 enzyme CYP3A4 primarily metabolizes it, so care must be taken when co-administering it with drugs that induce or inhibit CYP3A4 [7]. Also, patients with renal or hepatic dysfunction should adjust their doses [8]. The transdermal delivery of lurasidone can eliminate these limitations and be a potential alternative for oral delivery. The skin is an attractive site for drug delivery due to its extensive surface area and accessibility [9]. The skin, encompassing a surface area of about 1 to 2 square meters, is the body’s largest organ [10]. Transdermal drug delivery systems are designed to deliver active ingredients through the intact skin into the systemic circulation. However, the stratum corneum, the outermost layer of the skin, acts as a barrier that only allows small, lipophilic molecules to pass through. This layer contains both hydrophilic and hydrophobic regions, and prevents the entry of foreign substances [11]. Transdermal delivery offers several advantages, including maintaining a controlled amount of drug in plasma levels, reducing the risk of adverse effects, and eliminating hepatic first-pass metabolism [12]. Microneedle technology is a type of transdermal delivery system that offers a less invasive and controlled method of drug delivery [11]. Microneedles, characterized by their micron size needles, create micro-scale channels in the skin, effectively breaching the stratum corneum barrier without causing pain [9]. These channels enable the delivery of a wide array of therapeutic agents, ranging from small molecules to macromolecules and biologics, into skin layers. This capability facilitates both localized and systemic treatments [13]. Upon application to the skin, microneedles penetrate the epidermal layer and deliver compounds into the underlying dermis, which has high vascular and lymphatic circulation. This enables the systemic absorption of the drug, allowing for sustained and controlled release [14]. Microneedles offer several advantages over traditional hypodermic needles, making them a promising technology in the realm of drug delivery. These benefits include reduced invasiveness, an ease of administration, and enhanced patient acceptance [15]. Various types of microneedles exist, including hollow, biodegradable solid/dissolving, and other polymeric microneedles, which can be coated or loaded with the compound that is intended to be delivered [14]. Dissolvable microneedle patches, composed of water-soluble polymers, are designed for disintegration after being inserted into the skin. These patches are commonly made using water-soluble polymers like polyvinyl alcohol (PVA) and polyvinylpyrrolidone (PVP). Their high solubility means that they can quickly dissolve upon contact with the interstitial fluid in the skin and release their content into skin layers, thus facilitating the transdermal delivery of drugs [16]. Nanotechnology stands out as a rapidly expanding field, especially nanoparticles and nanomaterials [17]. Nanoparticles are small, solid, spherical structures that are made from either natural or synthetic polymers. PLGA stands out as one of the most widely used biodegradable polymers with extensive applications in polymeric nanoparticle formulation due to its attractive properties. Firstly, its biodegradability and biocompatibility ensure its safety, and it has gained approval from both the FDA and the European Medicine Agency for use in drug delivery systems. Moreover, it offers protection to drugs from degradation. PLGA nanoparticles’ diverse utility comes from their ability to carry various types of therapeutic agents such as small-molecule drugs, whether they are hydrophilic or hydrophobic, and big molecular drugs, vaccines, and complex biological macromolecules [18]. Nanoparticles play an important role in enhancing the bioavailability, solubility, and transdermal penetration of many drugs when compared to traditional topical formulations [19]. Additionally, they can enable the delivery of the desired amount of a drug over a prolonged period [20,21]. When nanoparticles are incorporated into microneedles, they synergize with the advantages of microneedle technology, enabling an even more effective delivery of therapeutic agents [22]. Nanoparticles can be efficiently delivered across the stratum corneum barrier by embedding them into the tips of microneedles that are then applied directly to the skin. Adding nanoparticles in microneedles can enable the delivery of both hydrophilic and hydrophobic drugs, which extends the scope of drugs that can be delivered transdermally [23]. Researchers have explored the potential of these microneedle patches, which can gradually release nanoparticles, providing prolonged therapeutic effects. This innovation holds significant promise for long-acting self-administration in chronic conditions, potentially improving patient compliance compared to conventional daily pills or injections [14].

Effervescent microneedle patches further elevate the possibilities of transdermal drug delivery [24]. Effervescent composition is a mixture of sodium bicarbonate and organic acids such as citric and tartaric acid [25]. By adding an effervescent formula in the patch’s backing, a reaction will occur between sodium bicarbonate and citric acid upon contact with the interstitial fluid in the skin. This reaction produces carbon dioxide bubbles, which aid in the weakening of the microneedles’ attachment to the patch, allowing them to separate from the skin in a few minutes after insertion [26]. The incorporation of effervescent components ensures that the microneedles remain in the skin after application. Moreover, these patches can be self-administered with patch wear times of a few minutes and yet achieve the sustained release of drug from the embedded microneedles in the skin [27]. This innovation adds another layer of control to the drug delivery process, enhancing its reliability and effectiveness [24].

The main aim of this study is to develop and evaluate a transdermal delivery system for lurasidone. This approach is designed with a dual purpose: firstly, to enhance the bioavailability of lurasidone, potentially reducing its side effect, and secondly, to increase patient adherence to the medication regimen. By considering the patch size, the oral dosage requirements, and the bioavailability percentage of lurasidone, our aim was to deliver 18 µg/cm^2^/day of the drug into the systemic circulation for a three-day period. The in vitro delivery of lurasidone with the help of chemical enhancers, Dr. Pen^TM^ Ultima A6, and microneedles was analyzed. We successfully achieved the transdermal delivery of a dose equivalent to 20 mg of oral lurasidone, using biodegradable microneedle patches loaded with lurasidone nanoparticles. These microneedles, fabricated with PVA and PVP polymers, detached from the patch with the use of an effervescent backing formula.

## 2. Materials and Methods

### 2.1. Materials

Lurasidone was procured from BOC SCIENCES (New York, NY, USA). PLGA (poly lactic acid (PLA) and polyglycolic acid) was sourced from ChemScene (Monmouth Junction, NJ, USA). Polyethylene glycol was purchased from Fisher Scientific (Pittsburgh, PA, USA), while Volpo^TM^ (Brij^®^O20: Polyoxyethylene (20) oleic ether) was acquired from Sigma Aldrich (St. Louis, MO, USA). A master stainless steel microneedle structure consisting of 10 × 10 arrays of 500 μm needles was procured from Micropoint Technologies Ltd. (Singapore). Polyvinyl Pyrrolidone (PVP), polyvinyl alcohol (PVA) 18-88, citric acid, and sodium bicarbonate were sourced from Sigma Aldrich (St. Louis, MO, USA). Transcutol, oleic acid, and isopropyl myristate were received as gift samples from Croda Inc. (Edison, NJ, USA). Porcine ear skin was obtained from Animal Technologies (Tyler, TX, USA). All high-performance liquid chromatography (HPLC) grade solvents were sourced from Fisher Scientific in Pittsburgh, PA, USA. Dr. Pen^TM^ was purchased from Amazon.

### 2.2. Methods

#### 2.2.1. Quantitative Analysis

To quantify the amount of lurasidone that penetrated through the skin, all samples were analyzed using the water alliance 2695 separations model high-pressure liquid chromatography (HPLC) system (Waters, MA, USA).

A reverse phase HPLC method was developed to quantify lurasidone. The samples were analyzed using a 250 mm × 4.6 mm Inertsil C18 ODS-3V column with a mobile phase combination of 10 mM sodium citrate buffer (pH 5.3) as the buffer phase and a mixture of acetonitrile and methanol in a 5:95 ratio as the organic phase. The injection volume was 10 µL, the flow rate was 1 mL/min, and the column temperature was maintained at 40 °C. The drug peak was observed at 6.8 min, and the total run time for the analysis was 10 min. The detection wavelength was 230 nm. The developed method was validated for accuracy, precision, linearity, limit of detection (LOD), and limit of quantification (LOQ) assessments.

#### 2.2.2. In Vitro Permeation Test (IVPT) Study

For in vitro permeation studies, dermatomed porcine ear skin was used. The skin was carefully removed from the porcine ear utilizing forceps and scissors. Following this, the full-thickness porcine skin was dermatomed using a Nouvag AG Dermatome 75 mm (Goldach, Switzerland). The thickness of the dermatomed porcine skin used in the study was 300–800 µm. Before the investigation, the skin was thawed using phosphate-buffered saline with pH 7.4 and then cut into smaller pieces [17]. In vitro permeation tests were conducted using vertical static Franz diffusion cells with 0.64 sq.cm of permeation area and a receptor volume of 5 mL (PermeGear, Inc., Hellertown, PA, USA). The water jacket enclosing the diffusion cells was attached to a water bath with circulation, ensuring that the temperature of the receptor chambers remained at 37 °C. Skin temperature was at 32 °C. 

The saturation solubility of lurasidone in propylene glycol and chemical enhancers, such as 25% *w*/*w* oleic acid, 5% *w*/*w* transcutol, and 35% *w*/*w* isopropyl myristate in propylene glycol, was examined. An excess amount of lurasidone was added to 1 mL of each enhancer, followed by overnight agitation on a platform shaker at ambient temperature. The mixtures were then centrifugated at 13,400 rpm for 15 min to separate the supernatant from the undissolved lurasidone. A 0.45 µm filter was employed to remove residual particulate matter from the supernatant.

##### Skin Integrity and Resistance Measurement

To ensure the skin’s structural integrity, the skin was mounted on Franz cells and its resistance was measured. The resistance of the skin was analyzed using silver/silver chloride electrodes, an Agilent 33220A 20 MHz function/arbitrary waveform generator, and an Agilent 34410A 6.5-digit multimeter (Agilent Technologies, Santa Clara, CA, USA) [28]. Skin pieces exhibiting resistance lower than 10 k cm^2^ were discarded from the experiment. The donor formulation was added after attaching the donor chamber to the mounted skin.

##### Lurasidone Detection in Receptor Fluid and Skin

A receptor fluid composed of 10 mM Citric acid at pH 3 was used to maintain sink skin conditions. Receptor samples (300 µL) were taken from the sampling arm at designated time intervals during a 72 h duration. The volume of the receptor was restored by adding 300 µL of fresh receptor medium. For analyzing the amount of lurasidone in the skin, after the 72 h study, the skin was removed from the Franz cells, and the area of skin exposed to permeation was cut into 3 pieces. The diced skins were put in a vial and 3 mL of acetonitrile was added to extract the drug from the skin. The skin vials were put on a platform shaker at 150 rpm overnight. After that, acetonitrile was collected from the skin vial and passed through 0.22 µm nylon membrane filters (CELLTREAR Scientific Products, Pepperell, MA, USA) to remove skin residue. The amount of lurasidone in all collected samples was analyzed using a validated HPLC method to determine the extent of lurasidone penetration through the skin barrier.

#### 2.2.3. Passive Delivery and Effect of Chemical Enhancers

The passive permeation and the effect of chemical enhancers on the delivery of lurasidone were evaluated in IVPT studies conducted on dermatomed pig ear skin. For the passive permeation, the donor solution consisted of 100 µL of 90% saturated solution of lurasidone in propylene glycol. In the case of the chemical enhancer group, the donor solutions included 90% saturated solution of lurasidone in 10% *w*/*w* oleic acid in propylene glycol, 5% *w*/*w* transcutol in propylene glycol, and 35% *w*/*w* isopropyl myristate in propylene glycol.

#### 2.2.4. Statistical Analysis

Experimental results were represented as the mean ± standard error of the mean (SEM) with a sample size of four (n = 4). All statistical analyses were conducted using Graph Pad Prism. One-way ANOVA (Analysis of variance) was utilized to discern significant differences among the groups. A *p*-value less than 0.05 was considered statistically significant.

#### 2.2.5. Preparation of PLGA Nanoparticles

Nanoparticles were prepared using the oil-in-water (O/W) emulsion solvent evaporation method. A 500 µL solution containing a 3:1 ratio of 25 mg PLGA to 8.3 mg of drug in dichloromethane (DCM) was added dropwise to 3 mL of 1% *w*/*v* PVA using an automatic pipette. At the same time, the mixture was homogenized at a speed of 10,000 rpm using a Bead Ruptor (Omni International, Atlanta, GA, USA). Microscopic evaluations were performed using a Leica DM 750 microscope at intervals of 5 min, extending up to 15 min, to analyze the solution. The study was also repeated using 2% *w*/*v* and 3% *w*/*v* PVA concentrations. In addition, the effects of homogenization at speeds of 24,000 rpm, 22,000 rpm, and 18,000 rpm for 1 min were analyzed. Also, the same 500 µL solution with a 3:1 ratio of PLGA to the drug in DCM was introduced into 3 mL of 1% *w*/*v* PVA using a syringe pump. The study was repeated using acetone, acetonitrile, and tetrahydrofuran (THF) in place of DCM to check the effect of different organic solvents on the size of nanoparticles. Two concentrations of PLGA, 100 mg/mL and 50 mg/mL were analyzed in an organic solution. Various PLGA-to-drug ratios were investigated. Different combinations of organic phase to aqueous phase were analyzed; for example, 200 µL of PLGA and drug solution in THF were incorporated into varying volumes of 1% *w*/*v* PVA—specifically 3, 4, 6, 7, and 8 mL. Table 1 shows the summary of nanoparticle optimization. The final solution underwent centrifugation at 4000 rpm for 10 min. An Eppendorf centrifuge 5810 R was used in this process (Eppendorf, Westbury, NY, USA). This step was repeated three times. The initial centrifugation aimed to remove PVA, while the subsequent three rounds served to wash the nanoparticles and eliminate any unencapsulated drug. After each centrifugation, the supernatant was replaced with an equivalent volume of DI water. Following the final wash, the supernatant was discarded, and the nanoparticles were transferred to a vial. A few milliliters of 2% *w/v* trehalose were added to the nanoparticles to protect them from cold. Thereafter, this vial was pre-frozen at −80 °C for two hours, followed by lyophilization for 48 h.

Two variables were investigated in the preparation of nanoparticles. The first variable encompassed the choice of organic solvent, where acetone and THF were used. The second variable involved the method of preparation, evaluating both homogenization and syringe pump techniques.

#### 2.2.6. Characterizations of Nanoparticles

##### Nanoparticle Size

After nanoparticle formulation, particle size and polydispersity index were observed using a Malvern instruments ZEN3600 Zeta Sizer (Mal 1096674) (Malvern Panalytical Ltd., Malvern, UK). A mixture was prepared by combining 0.5 mL of the dense nanoparticle solution with 0.5 mL of deionized water. To determine the particle size, each specimen underwent six consecutive measurements [29].

##### Entrapment Efficacy

The concentration of lurasidone in the PLGA nanoparticles was analyzed by weighing 5 mg of nanoparticles, followed by the addition of 0.2 mL of acetonitrile and 0.8 mL methanol. The nanoparticles were solubilized in acetonitrile through vortex mixing. The subsequent addition of methanol induced the precipitation of the PLGA polymer, while lurasidone remained soluble in the solvent mixture of acetonitrile and methanol. After this, the mixed solution underwent centrifugation at 13,400 rpm for 15 min. The overall content of lurasidone in 5 mg nanoparticles was then quantified utilizing HPLC.

#### 2.2.7. Release Studies

To determine whether the final formulation of nanoparticles (3:1 PLGA/drug ratio) releases drugs upon contact with water, a release study was performed. For the release study,10 mg of the nanoparticles was added to a vial with 1 mL of citric acid solution (pH 3) on a magnetic stirrer with the speed of 500 rpm for an analysis period of 72 h at ambient temperature. At predetermined intervals, 300 µL of the solution was withdrawn from the vial, and an equivalent volume of fresh citric acid solution was added back. Each sample was filtered with a 0.22 µm filter to separate nanoparticles. For the detection of free lurasidone in the solution, the sample was then analyzed using HPLC.

#### 2.2.8. Microneedle Treatment Dr. Pen^TM^ Ultima A6

The Dr. Pen^TM^ Ultima A6, a device used in the cosmetic industry, works as an automated microneedle system that has streamlined wireless. It uses sterile, sealed, disposable needle cartilage to create micro channels in the skin [30]. For this study, Dr. Pen^TM^ Ultima A6 was applied to the skin for 10 s, using a 500 micrometer needle-length setting. Subsequently, two preparations—lurasidone in propylene glycol and lurasidone nanoparticles in propylene glycol—were applied to microneedle-treated skin on a Franz cell setup. In addition, lurasidone nanoparticles in propylene glycol were applied to untreated skin for comparative analysis. Each Franz cell was loaded with 100 microliters of solution. Samples were taken for 72 h.

#### 2.2.9. Poly Dimethyl Siloxane (PDMS) Molds

The Sylgard^®^186 silicone elastomer base and curing agent were accurately weighed and blended in a 10:1 weight-to-weight ratio. Two methods were used to make molds. First, this pre-mixed PDMS mixture was carefully poured into a 6-well plate containing distinct master structures. To eliminate air bubbles, the filled container was placed in a vacuum drying oven and subjected to a vacuum of 200 mbar at 25 °C for 15 min [31]. Next, the premixed PDMS mixture was added to a centrifuge tube containing the master structure and centrifuged for 10 min at 4000 rpm to remove all bubbles. Afterward, the molds that were made using both methods underwent a curing process at 60 °C for 4 h. Once cured, the master structures were separated from the molds.

#### 2.2.10. Microneedle Preparation

##### Fabrication of Tip-Loaded Effervescent Microneedle Patch

PVA is a biocompatible polymer with water solubility and chemical resistance [32]. PVP is a polymer that is soluble in water and capable of absorbing up to 40% of its weight in water. It forms films when dissolved [33]. A blend of PVP and PVA for microneedles offers advantages like biocompatibility, controlled drug release, and mechanical strength. These polymers ensure that microneedles safely dissolve in the skin. Two groups of microneedles were fabricated: one group was filled with a suspension of lurasidone in PVA and PVP, while the other group contained lurasidone nanoparticles.

For the first group, the optimal mix of material for maximum drug solubility in the microneedles was determined by evaluating the saturation solubility of lurasidone in 1% *w*/*v* PVA and varying percentages of PVP using HPLC. An excess of lurasidone was introduced to 1 mL of each solvent mixture, which was then placed on a platform shaker overnight for thorough mixing. Post shaking, the combinations underwent centrifugation at 13,400 rpm for 15 min. The resultant supernatants were carefully collected and passed through a 0.22-micrometer filter for purification. Finally, the filtered supernatants were analyzed using HPLC. To further increase the drug amount in the microneedle formulation, a suspension of the drug was made in the mixture of PVA and different percentages of PVP. The homogenization technique was applied to decrease the drug particles in the mixture and increase suspension stability. A mixture was prepared by adding 74 mg of lurasidone to 3.5 mL of a solution composed of a 1:2 ratio of 1% *w*/*v* PVA to 20% *w*/*v* PVP, and 5% *w*/*w* transcutol HP. This mixture was then homogenized at 24,000 rpm for 3 min, followed by 18,000 rpm for 2 min. For the second group, nanoparticles combined with 1% *w*/*v* PVA were further analyzed in conjunction with three separate aqueous solutions of PVP at 20%, 15%, and 10% *w/v* concentrations. Two solutions using PVP of molecular weights 360,000 and 40,000 for each concentration were prepared to analyze their effect on the final microneedle. Four distinct ratios were studied. This analysis aimed to evaluate the mechanical strength of the microneedles. The same procedure was repeated with lurasidone powder. Table 2 is the summary of all combinations.

The previously prepared PDMS molds were loaded with 80 µL of the drug solution. The molds underwent centrifugation at 4000 rpm for 10 min to ensure uniform distribution. Also, a vacuum treatment was applied using a vacuum drying oven at 200 mbar and 25 °C for 6 min. Moreover, a combination of both methods was employed to analyze their effect on the final microneedle. Subsequently, the mold containing the solution was placed in an oven and maintained at 60 °C overnight to facilitate drying.

##### Effervescence for Backing Part

A mixture of citric acid (4% *w*/*w*) and sodium bicarbonate (5% *w*/*w*) were prepared in ethyl acetate [25]. During optimization, three different ratios of citric acid to sodium bicarbonate, 4:5, 8:10, and 12:15, were evaluated in 10 mL of ethyl acetate. Next, the effervescent solution was dispensed onto the molds, which already contained the dried drug solution in their tips. Since the effervescent solution was brittle after drying in the mold, 10% *w*/*v* PVP was added to increase its hardness.

#### 2.2.11. Microneedle Characterization

##### Parafilm M^®^ Microneedle Insertion Analysis

Parafilm M^®^ was used as a skin mimetic substrate for in vitro insertion tests. The penetration capacity of various microneedle mixtures was evaluated using four parafilm layers, each having an average thickness of 0.124 ± 0.001 mm. After application, each Parafilm M^®^ layer was separated by hand and analyzed using a Leica DM 750 optical microscope. If the needles successfully penetrated the third Parafilm M^®^ layer, their hardness was considered sufficient for further analysis [34].

##### Drug Analysis in Microneedle Tips

To analyze the overall lurasidone amount in the microneedle, the needles of the microneedle were carefully removed from the base using a scalpel. These dissected tips were subsequently moved to a 20 mL vial. To dissolve the PVA and PVP components in the microneedles, 2 mL of water was added to the vial and then placed in a shaker for 3 h. After this, 5 mL of acetonitrile was added to solubilize lurasidone, and the mixture was shaken for an additional 3 h. The final solution was then analyzed using HPLC [27].

##### Microneedle Insertion on the Skin

Using the Stable Micro Systems TA.XT Express Texture Analyzer (Texture Technologies Corp., South Hamilton, MA, USA), the microneedles were inserted into the skin. For this procedure, the skin was positioned on the analyzer’s base plate and microneedles were placed on the top of skin, ensuring the microneedles faced the skin. The Texture Analyzer featured a 7 mm diameter stainless steel probe. The operation was conducted in compression mode, where the steel probe moved downward at a consistent speed of 1 mm/s to press the microneedles into the skin. This compression was maintained for 2 min before the probe was retracted. The microneedles were left in contact with the skin for 10 min before being carefully removed.

##### Histology Study

Histological studies were conducted for confirmation to assess the microchannels formed by microneedles [35]. The untreated porcine ear skin and the porcine ear skin treated with lurasidone microneedles were put in a flat position in a Tissue-Tek^®^ optical coherence tomography (OCT) compound medium. These skin samples were oriented with the stratum corneum facing upward and frozen at −80 °C for at least 1 h to solidify before sectioning. Sectioning was carried out using a Leica CM1860 cryostat. The resulting skin sections were 10 µm in thickness and were stained with hematoxylin and eosin. These stained cross-sectional slices were then mounted onto glass slides from Globe Scientific, Inc. (Paramus, NJ, USA) and were observed and photographed using a Leica DM 750 microscope.

##### Mechanical Strength of Microneedles

The mechanical strength of microneedles is important in ensuring that they remain intact under the force exerted during insertion. To ascertain the microneedles’ mechanical strength, we utilized the Stable Micro Systems TA.XT Express Texture Analyzer from Texture Technologies Corp., South Hamilton, MA, USA. In the setup, a microneedle patch measuring 1 cm by 1 cm was positioned on the analyzer’s base plate, with the needle tips oriented upwards. The evaluation was conducted in compression mode. The steel probe moved at a velocity of 1.00 mm/s with an initial contact force of 5 g. Following this, the probe retracted at a speed of 5 mm/s, covering a test distance of 1 mm. 

#### 2.2.12. SEM (Scanning Electron Microscopy) Analysis

The characteristics of the microneedles were studied using a JEOL JSM-IT700HR Scanning Electron Microscope (SEM; JEOL, Tokyo, Japan), equipped with a Schottky-type, in-lens thermal field emission gun (FEG). Samples were mounted on SEM stubs and loaded in the SEM without any metal coating to be imaged at an accelerating voltage of 2.3 kV. SEM micrographs were acquired using an Everhart Thornley-type secondary electron detector at various magnifications as indicated. ImageJ was used to measure length, width, and needle-to-needle distance (National Institute of Health, Bethesda, MD, USA). Three replicates were used for accuracy.

#### 2.2.13. IVPT Study with Microneedles

An IVPT study evaluated the microneedle patch with an effervescent backing membrane. First, the microneedle was inserted into the skin using the Stable Micro Systems TA.XT Express Texture Analyzer. The steel probe moved at a velocity of 1.00 mm/s with an initial contact force of 5 g and was held on the microneedles for two minutes. Following this, the skin was mounted on a Franz diffusion cell. After a 10 min interval, the effervescent backing was removed, leaving only the inserted microneedle in the skin. The study was conducted over a duration of 72 h and comprised two distinct groups: one group utilized a microneedle loaded with lurasidone nanoparticles, while the other group used a microneedle containing only lurasidone.

## 3. Results

### 3.1. Quantitative Analysis

The established HPLC technique was tested for consistency across different days (inter-day) and within the same day (intra-day) for its accuracy, precision, and linearity. Intra-day and inter-day precision and accuracy were evaluated by examining triplicate sets over three distinct concentrations. This technique identified a detection threshold of 0.02 µg/mL, and a quantification threshold of 0.05 µg/mL. Figure 1 shows the chromatogram for lurasidone at a 50 µg/mL concentration in acetonitrile.

### 3.2. In Vitro Permeation Test (IVPT) Study

Different solvents were examined to identify the most suitable receptor solution, ensuring sink conditions for lurasidone entering the receptor. Table 3 shows the solvents and respective solubilization capacities for lurasidone. Polyethylene glycol (PEG) demonstrated the highest solubility for lurasidone. However, due to its high viscosity, an 80% *v*/*v* PEG and 20% *v*/*v* Volpo 6% *w*/*v* mixture was initially selected as the receptor solution. Following in vitro permeation tests (passive delivery, delivery with chemical enhancers, and the use of Dr. Pen^TM^), only trace amounts of lurasidone were detected in the skin, with no penetration into the receptor. It was observed that the PEG component in the solvent induced skin dryness before the completion of the first day, so it hindered drug penetration. Therefore, this mixture proved unsuitable for a 3-day study duration. Therefore, the solubility of lurasidone was tested in citric acid buffers of varying pH levels to identify an alternative receptor solution. The receptor solvent was changed to a 10 mM citric acid buffer with a pH of 3.

### 3.3. Passive Testing

The saturation solubility of lurasidone in propylene glycol was 954.55 ± 34 µg/mL. For 72 h in vitro evaluations, a concentration equivalent to 90% of the saturation solubility, 859.1 ± 31 µg/mL, was employed. Notably, no penetration was observed in the receptor.

### 3.4. Delivery with Chemical Enhancers

The determined saturation solubility of lurasidone in various solvent systems was as follows: 1603.94 ± 9 µg/mL in a mixture of 25% *w/w* oleic acid in propylene glycol, 1351 ± 50 µg/mL in 35% *w*/*w* isopropyl myristate in propylene glycol, and a significantly higher 5005.5 ± 50 µg/mL in 5% *w*/*w* transcutol in propylene glycol. In the IVPT, solutions at 90% of their saturation solubility were utilized, and 100 µL of each solvent was added to the donor. After completing a 72 h in vitro study, the delivery profiles for each formulation were plotted. As is shown in Figure 2, the transcutol group exhibited the highest delivery, with a value of 65.6 ± 11.88 µg/cm^2^. This was followed by the isopropyl myristate group, which delivered 51.6 ± 16.74 µg/cm^2^. The oleic acid group demonstrated the lowest delivery, which was 10.07 ± 1.09 µg/cm^2^. None of these delivery values met the desired target. 

### 3.5. Fabrication of Lurasidone Nanoparticles 

Initially, a PLGA-to-drug ratio of 3:1 (30 mg of PLGA and 10 mg of drug) was chosen for optimization. DCM served as the initial organic solvent, with a 1% *w*/*v* PVA solution selected as the aqueous phase. To formulate nanoparticles, a comparative study used homogenization at varied rpm and time intervals to prepare nanoparticles. Based on microscopic observations, neither approach initially resulted in acceptable desired size and encapsulation attributes for nanoparticles.

Next, 2% *w*/*v* and 3% *w*/*v* PVA were investigated to discern the impact of PVA percentage on nanoparticle formation. Based on microscopic observation, the 1% *w*/*v* PVA solution generated more nanoparticles. Two PLGA concentrations in organic solvent were evaluated: 100 mg/mL and 50 mg/mL. A higher concentration of PLGA resulted in larger nanoparticle sizes. Also, a 100 mg/mL concentration needed an overnight dissolution and yet not all PLGA dissolved, even after extended shaking. Thus, the 50 mg/mL concentration was deemed more suitable. The organic solvent was then substituted with acetonitrile, acetone, and THF. Acetonitrile yielded the smallest nanoparticles, followed by acetone, while THF produced the largest. Based on a previous study, it was found that having a range of sizes for PLGA nanoparticles was essential for creating a dense packing in both bilayer conical and pyramidal microneedles [36]. Since varied sizes of nanoparticle was the goal, acetone and THF were chosen for further analysis. Nanoparticles were made using two methods to better compare the homogenization and syringe pump techniques, as well as the use of acetone versus THF.

Initially, a 500 µL solution of 30 mg of PLGA and 10 mg of drug (3:1) in acetone was combined with 3 mL of 1% *w*/*v* PVA using homogenization. Subsequently, the same 500 µL 1:3 drug/PLGA solution in acetone was added into 3 mL of 1% *w*/*v* PVA using a syringe pump. These procedures were then repeated, using THF instead of acetone as the organic solvent.

Dynamic light scattering, utilizing a Zeta sizer, revealed distinct differences in nanoparticle characteristics. As illustrated in Table 4, the nanoparticles produced with THF and the syringe pump exhibited the largest size and the highest polydispersity index (PDI), followed by the nanoparticles made using THF and homogenization. In contrast, the nanoparticles created with acetone via the syringe pump method demonstrated the smallest size. Since the objective was to aim for higher polydispersity to produce densely packed microneedles and based on the drug’s recovery rate and its encapsulation efficiency, which were analyzed using HPLC, the nanoparticles that were made with THF using the syringe pump were selected for future studies.

To identify the optimal drug-to-PLGA nanoparticle formulation based on encapsulation and size, 400 µL of various combinations of PLGA and drug was prepared in THF and then added to 8 mL of 1% *w*/*v* PVA solution. Following this, Zeta sizer and HPLC were used to evaluate encapsulation efficiency, PDI, and particle size. In this study, we were aiming for higher encapsulation. As detailed in Table 5, the combinations of 50 mg of PLGA with 10 mg of drug (5:1) and 30 mg of PLGA with 10 mg of drug (3:1) yielded the most promising results out of the tested combinations. 

Subsequent experiments explored different proportions of the organic phase to the aqueous phase. For the chosen ratios, 200 µL samples in THF were mixed with varying volumes of 1% *w/v* PVA, including 3, 4, 6, 7, and 8 mL.

It was observed that mixing 200 µL samples of both the 5:1 and 3:1 ratio with 8 mL of 1% *w/v* PVA led to aggregation. However, the mixture of a 3:1 PLGA/drug ratio with 3 mL of 1% *w/v* PVA demonstrates the highest encapsulation efficiency of 56%, a PDI value of 0.416 (due to varied sizes), and a particle size measuring approximately 547.86 ± 11 nm. As illustrated in Figure 3, 3:1 PLGA/drug ratio nanoparticle shapes are round and vary in size. The debris present around the nanoparticles is trehalose which is used as a cryoprotectant during lyophilization.

### 3.6. Release Study

Over a 72 h sampling period, an increase in the concentration of lurasidone in the sample was observed starting from the 6 h mark, indicating that the nanoparticles began releasing the drug at this point. Some amount of lurasidone was detected at 0 h sampling, which is the result of free lurasidone on the surface of the nanoparticles.

### 3.7. IVPT with Dr. Pen^TM^ Ultima A6

An IVPT was conducted on three distinct groups, each consisting of four replicates. Skin in two of the groups was pretreated with the Dr. Pen^TM^ Ultima A6 device, while the third group utilized intact, untreated skin. The drug concentration and drug content were quantified as 721.33 µg/mL and 112.71 µg/cm^2^ for lurasidone nanoparticles in propylene glycol and 917 µg/mL and 143.28 µg/cm^2^ for lurasidone in propylene glycol, respectively. The study extended over a 72 h duration.

As depicted in Figure 4, all three groups exhibited very low delivery to the receptor. An explanation for this could be lurasidone’s hydrophobic nature, which prevented it from effectively passing through the hydrophilic channels that were created by Dr. Pen^TM^ Ultima A6. Additionally, the size of the nanoparticles could be the reason for their low penetration to the receptor. Their dimensions are larger than the microchannels generated by the Dr. Pen^TM^ Ultima A6, limiting the nanoparticles to pass through hair follicles primarily.

### 3.8. Volume of Microneedle Tips

Calculations were conducted to estimate the volume of the drug solution that can transfer to microneedle tips in a 50 cm^2^ microneedle patch. 

The tip of each microneedle is architecturally pyramidal. The base of this pyramid is a square with dimensions of 150 micrometers by 150 micrometers, and the height is quantified as 500 micrometers.

To determine the volume of such a pyramidal structure, the following formula is used:*V* = 1/3 × *b* × *h* = 1/3 × 150 × 150 × 500 = 3,750,000 µm^3^

Each µm^3^ is equal to 10^−15^ L^3^, so it will be 0.375 × 10^−8^ L = 0.375 × 10^–2^ µL.

Since there are 100 needles in 0.25 cm^2^, the 0.25 cm^2^ area can accommodate a drug solution volume of 0.375 µL, and therefore, a 50 cm^2^ microneedle patch can adjust a volume of 75 µL for a drug solution.

### 3.9. Preparation of Microneedles

#### 3.9.1. Lurasidone Concentration in Microneedle Solution

Given that lurasidone exhibits pronounced lipophilicity, it is anticipated that a considerable proportion of the drug will remain within the skin layers. Thus, only a limited quantity of lurasidone may reach the receptors. This underscores the necessity to load the needle tips with a drug quantity exceeding the targeted dose. The solubility of lurasidone in PVA and PVP was assessed to determine if it could achieve the target dose over three days. As illustrated in Table 6, none of the vehicles could accommodate enough drugs to achieve the therapeutic dose for 3 days.

Consequently, due to the insufficient solubility of lurasidone, a suspension formulation of lurasidone was prepared. To further enhance the amount of lurasidone in the solution, 5% *w*/*w* transcutol was incorporated. Transcutol was selected based on its proven efficacy in improving lurasidone delivery, as observed in our studies comparing various chemical enhancers (the results can be found in Section 3.4).

Moreover, homogenization was employed to improve the homogeneity of the suspension and increase its concentration of lurasidone in the formulation. After aiming for suspension, using homogenization, and incorporating transcutol in formulation, lurasidone concentration increased and was determined to be 738.6 mg/mL in the solution composed of PVA and PVP. Figure 5 illustrates the impact of homogenization on the lurasidone particles within the solvent.

Both vacuum and centrifugation methods were used to fabricate microneedles. Additionally, a hybrid approach was tested initially using a centrifuge followed by exposure to a vacuum oven. Two different molecular weights of PVP (360,000 and 40,000) for each concentration were prepared to analyze their effect on the final microneedle. While none of the techniques that contained the centrifuge method produced strong needles, the vacuum process proved to be quite successful. Various blends like 1:2 (1% *w*/*v* PVA to 10% PVP), 1:2 (with 20% PVP), 1:1.5 (with 10% PVP), 1.5:1 (with 10% PVP), 1.5:1 (with 20% *w*/*v* PVP), and 1:1 (1% *w*/*v* PVA to 15% *w*/*v* PVP) managed to pierce as deep as the third layer of Parafilm M^®^. No significant differences were observed between the microneedles prepared using PVP of different molecular weights. After quantifying the lurasidone content in each sample using HPLC, a 1:1 (1% *w*/*v* PVA to 15% *w*/*v* PVP) ratio for nanoparticles and a 1:2 (1% PVA to 20% *w*/*v* PVP) ratio for pure lurasidone in microneedles were selected for the IVPT.

#### 3.9.2. Effervescent Backing Membrane

Effervescent backing is used to enable the quick and easy separation of the microneedles from the patch. The difference between a microneedle with an effervescent backing and a normal microneedle lies primarily in the ease of microneedle detachment from the skin. This reduces the time the microneedle patch needs to stay on the skin. Citric acid (4% *w*/*w*) and sodium bicarbonate (5% *w*/*w*) were dispersed in ethanol with varying concentrations of PVP [26]. Notably, as the concentration of PVP decreased, there was an increase in bubble production, which was desirable for this study. Combining 400 mg of citric acid with 500 mg of sodium bicarbonate (4:5), dissolved in 10 mL of 5% *w*/*v* PVP in ethanol, was found to be the most effective combination in detaching the blank microneedle base from the needles. This specific ratio facilitated the detachment of the needle from its backing.

After the homogenization of the drug-loaded microneedle formulation, it was observed that following the addition of the effervescent solution on the already dried microneedles’ tip, the microneedles began to dissolve into the backing layer which had ethanol, even though they were dried. A revised approach was taken where the same concentrations of citric acid and sodium bicarbonate were suspended in ethyl acetate. The new formulation showed brittleness after rapid drying but the microneedles remained in their position and did not dissolve back to the backing solution. To enhance the firmness of the backing, 10% *w*/*v* PVP in ethanol was added after the drying process. Figure 6 illustrates an effervescent microneedle patch as captured through a Wi-Fi digital microscope. This advanced electronic microscope features a built-in Wi-Fi hotspot, enabling seamless connectivity with iOS and Android smartphones and tablets for real-time image sharing and analysis.

### 3.10. Microneedle Characterization

#### 3.10.1. Histology Study

Histological analyses were conducted on both intact skin and skin treated with microneedles. As illustrated in Figure 7, the tips of the microneedles were able to penetrate the stratum corneum. To enhance the visibility of the microneedles under the microscope during the histology study, 10 µL of Nile red was incorporated into the formulation of the microneedles.

#### 3.10.2. Mechanical Strength of Microneedles

The mechanical strength of the microneedles was assessed by exerting an axial load on the microneedle array with the help of a Texture Analyzer. For successful skin penetration, the microneedle’s breaking force should surpass the required insertion force [24]. The average resistance at peak axial loading was recorded at 406.6 ± 3.6 g for a set of 100 microneedles. Given that the necessary force for microneedle insertion was 85.6 g, we can confidently state that the microneedles will remain intact during the penetration process.

#### 3.10.3. Mechanical Uniformity of Microneedles

The mechanical consistency of the microneedles was examined using four layers of Parafilm M^®^. As illustrated in Figure 8, all 100 microneedles managed to penetrate the third layer of Parafilm M^®^. The resulting perforations in Parafilm M^®^ took on a square shape, mirroring the pyramidal design of the microneedle tips.

#### 3.10.4. SEM (Scanning Electron Microscopy) Analysis

Scanning electron microscopy (SEM) was used for examining the length, width, needle-to-needle distance, and the configuration of microneedles, particularly focusing on their form. Based on ImageJ measurements, the length, width, and needle-to-needle distance were 305.82 ± 4.5 µm, 105.4 ± 3.2 µm, and 276.2 ± 6.8 µm, respectively. As is shown in Figure 9, lurasidone microneedles and lurasidone nanoparticles in microneedles both have pyramidal shape.

### 3.11. IVPT Study with Microneedles 

An IVPT was conducted using Franz diffusion cells. This study involved two groups, each with four replicates: one group with lurasidone in the microneedles and another with lurasidone nanoparticles in the microneedles. The experiment continued for 72 h. The drug concentration was determined to be 1282.17 μg/mL for lurasidone in the microneedle group and 1183 μg/mL for the lurasidone nanoparticles in microneedles. Regarding drug content, the microneedles with lurasidone held 512.87 micrograms in 0.25 cm^2^, while those with lurasidone nanoparticles contained 473.2 micrograms in 0.25 cm^2^. The microneedles had a permeation area of 0.25 cm^2^. As illustrated in Figure 10, the first group (lurasidone in microneedles) achieved an average delivery of 28.6 ± 4.5 micrograms, whereas the second group (lurasidone nanoparticles in microneedles) delivered approximately 55.49 ± 13.2 micrograms. Notably, the microneedles incorporating lurasidone nanoparticles demonstrated the potential to match the therapeutic dose of a 20 mg oral dose of lurasidone when applied via a 50 cm^2^ patch.

## 4. Discussion

The current study primarily focused on the development of an effervescent microneedle patch aimed at the systemic release of lurasidone in three days for the treatment of schizophrenia. Lurasidone is a lipophilic molecule (logP 4.56) that has the potential to be localized in the lipid layers of the skin, limiting its receptor delivery [35]. This was evident by the passive delivery of the drug dissolved in PG, where no receptor delivery was obtained. However, the incorporation of chemical enhancers enabled the receptor delivery of lurasidone. These chemical enhancers interact with the lipid layers in the skin and alter the microenvironment, thus facilitating the diffusion of the drugs. Prior research has indicated that there is a connection between the most appropriate lipophilicity of an enhancer and the lipophilicity of a drug. Therefore, chemical enhancers with different lipophilicity were chosen for delivering lurasidone [37]. As a result, transcutol, isopropyl myristate, and oleic acid were selected to evaluate their effect on the in vitro permeation of lurasidone. transcutol, which had the highest solubility of the drug, yielded the highest delivery. This could be because enhanced drug solubility in the vehicle modifies the thermodynamic driving force and improves penetration. Furthermore, transcutol has been demonstrated to reversibly permeabilize the skin barrier, thereby enhancing the solubility of drugs within the stratum corneum and facilitating its partitioning into and across the skin [38]. Transcutol was further utilized in the formulation of effervescent microneedles. In transdermal patch development, compounds with a low molecular weight (below 500 Da) and a moderate partition coefficient (log P) between 1 and 3 are considered suitable to deliver via a transdermal patch [39]. However, the molecular weight and logP of lurasidone are 492.68 g/mol and 4.5, respectively, and given the constraints that there is a limitation in the percentage of liquid (drug in PG with 5% *w*/*w* transcutol) that we can incorporate in the patch formulation, we concluded that it would be a significant challenge to design a patch to reach the target based on this formulation.

Later focus on this study then shifted to loading nanoparticles in the tips of effervescent microneedle patches. PLGA was the polymer of choice for formulating nanoparticles as it is more widely investigated and extensively characterized for providing tailored drug release [40]. Initial nanoparticle formulation attempts faced challenges, emphasizing the importance of optimizing solvents and processes. THF, which resulted in a higher polydispersity index, was selected. We also loaded the micronized suspension of lurasidone in the effervescent microneedles, with the addition of transcutol to the formulation to enhance its solubility further. This type of microneedle utilizes effervescent backing to facilitate the rapid separation of the microneedles from the patch upon contact with interstitial fluid in the skin. These systems have the ability to enable long-term drug dosing because the microneedle tips form a drug depot. W. Li et al. formulated effervescent microneedle patches for a lipophilic molecule, levonorgestrel, that achieved a month-long delivery [26].

In addition to this, the use of polymers that slowly degrade in the skin, thereby releasing the drug through slow dissolution in the dermis and into the systemic circulation, is one promising strategy for long-acting release. This study uses a combination of PVA and PVP polymers, which are known for fabricating a microneedle that is mechanically strong while maintaining a rapid rate of dissolution in skin fluids. This facilitates the swift exposure of drug particles to dissolution [34]. The mechanical integrity was further confirmed by using a skin simulant Parafilm M^®^ model, histology, and texture analyzer. The IVPT study with microneedles revealed the potential of lurasidone nanoparticle-loaded microneedles as an alternative way to oral delivery.

## 5. Conclusions

In our study, we aimed to achieve a targeted dose equivalent to a 20 mg oral dosage form of lurasidone. Different chemical enhancers were analyzed to deliver lurasidone to the receptor but none of them was able to deliver enough drug in order to reach the target. We used the Dr. Pen™ micro-needling device to determine the feasibility of lurasidone transport through microchannels created in the skin. The results for transporting lurasidone, either as a free drug or encapsulated nanoparticles, was not enough for the desired efficacy. Microneedle arrays with separatable effervescent backing, made of a combination of PVA and PVP, loaded with micronized lurasidone and PLGA nanoparticles of lurasidone, were successfully developed. The in vitro permeation test of delivering lurasidone via microneedles loaded with lurasidone nanoparticles was able to reach the projected therapeutic target for lurasidone. This suggests that employing nanoparticles can significantly enhance the therapeutic efficacy of microneedle-based delivery systems. The potential of achieving therapeutic levels with a 50 cm^2^ patch provides an encouraging outlook for the transdermal delivery of lurasidone, potentially offering a promising alternative to oral administration.

## Figures and Tables

**Figure 1 pharmaceutics-16-00308-f001:**
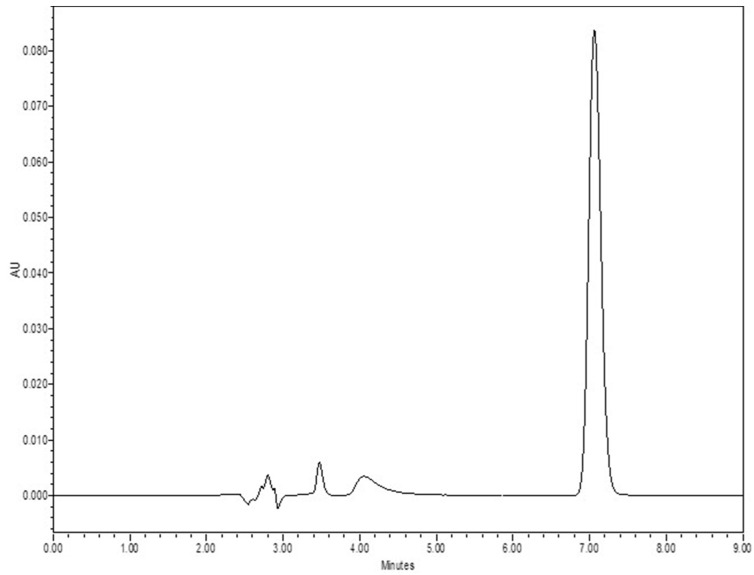
Chromatogram for LDN (lurasidone) at a concentration of 50 µg/mL in acetonitrile.

**Figure 2 pharmaceutics-16-00308-f002:**
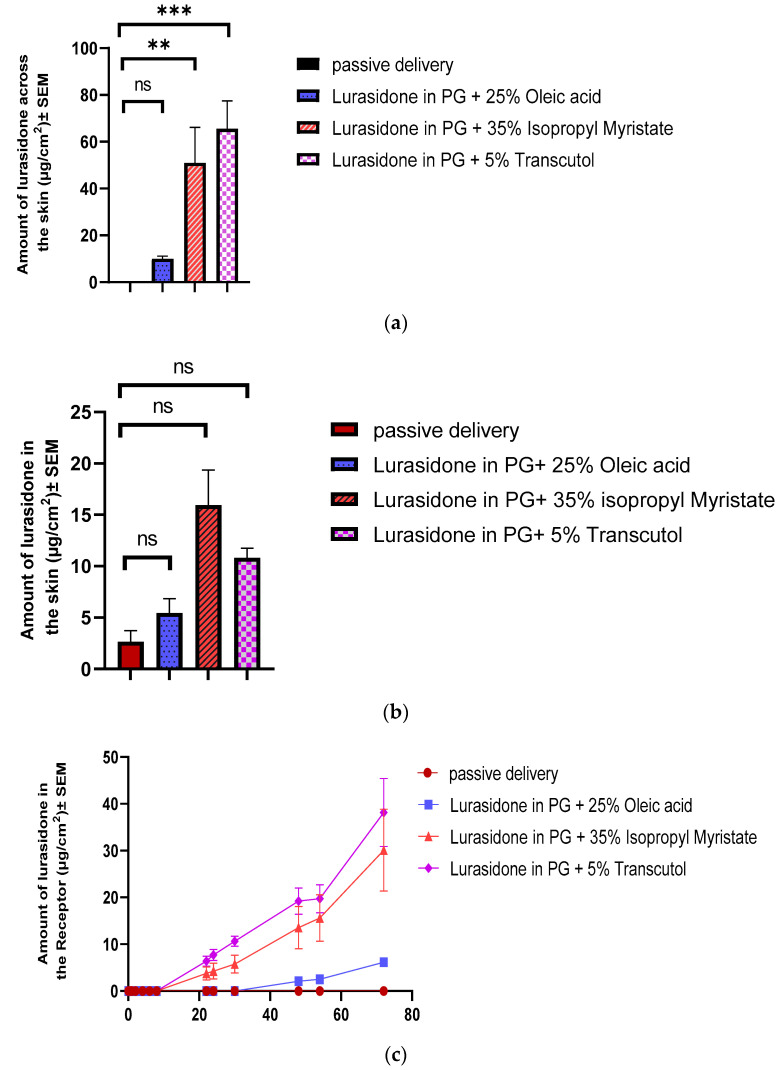
Evaluation of chemical enhancers for lurasidone delivery: (**a**) cumulative amount of lurasidone delivered into the receptor after 3 days; (**b**) amount of lurasidone delivered into the skin at the end of 3 days; and (**c**) permeation profile of lurasidone. Statistical analysis performed with one-way ANOVA; ns indicates no significant difference between groups, ** and *** indicate significant difference between groups, *p* < 0.05.

**Figure 3 pharmaceutics-16-00308-f003:**
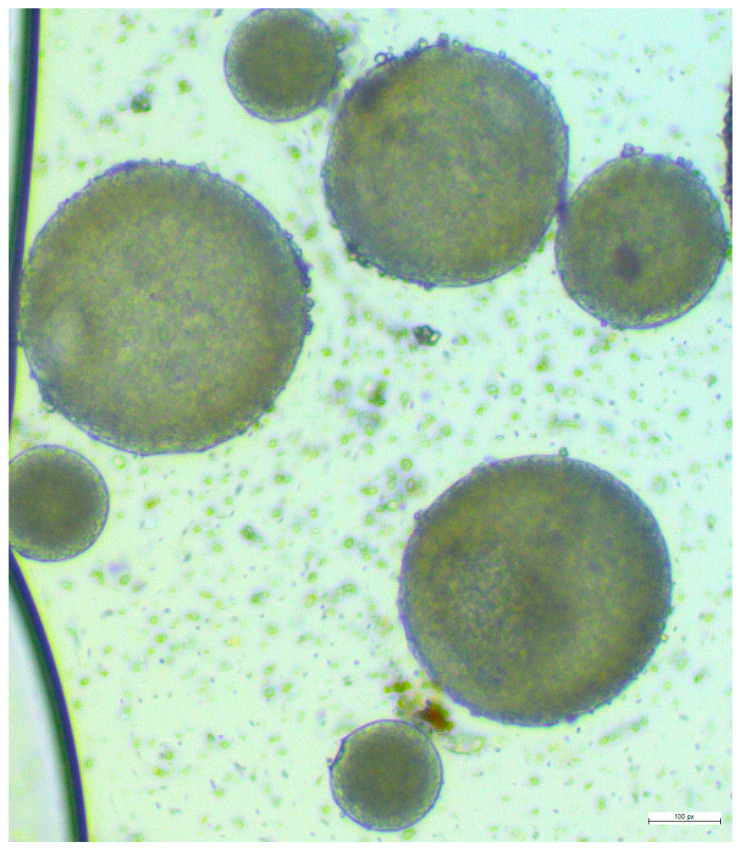
Microscopic view of the final formulation of lurasidone nanoparticles.

**Figure 4 pharmaceutics-16-00308-f004:**
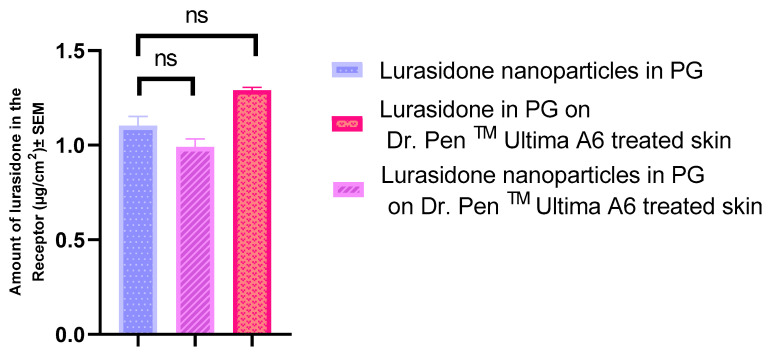
Evaluating the drug quantities that permeated through the skin to the receptor following Dr. Pen^TM^’s treatment. Statistical analysis performed with one-way ANOVA; ns indicates no significant difference between groups, *p* < 0.05.

**Figure 5 pharmaceutics-16-00308-f005:**
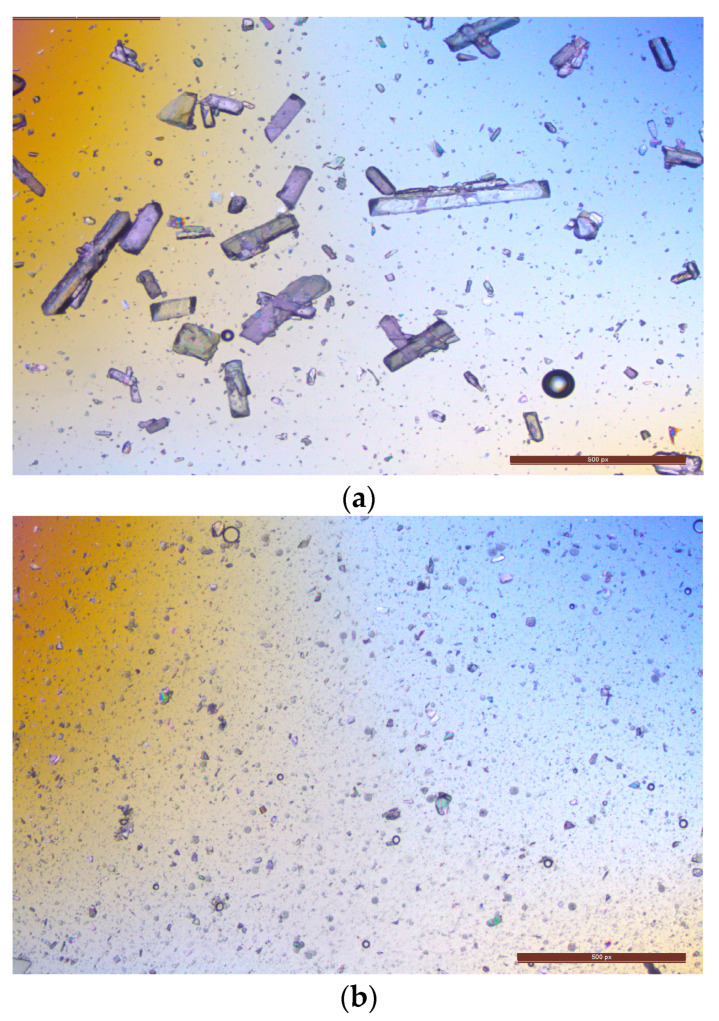
Analyzing the effect of homogenization on lurasidone particle size: (**a**) microscopic image of lurasidone in a 95% of 1:2 1% *w*/*v* PVA to 20% *w*/*v* PVP with 5% *w*/*w* transcutol; (**b**) microscopic image of the same suspension after 5 min of homogenization.

**Figure 6 pharmaceutics-16-00308-f006:**
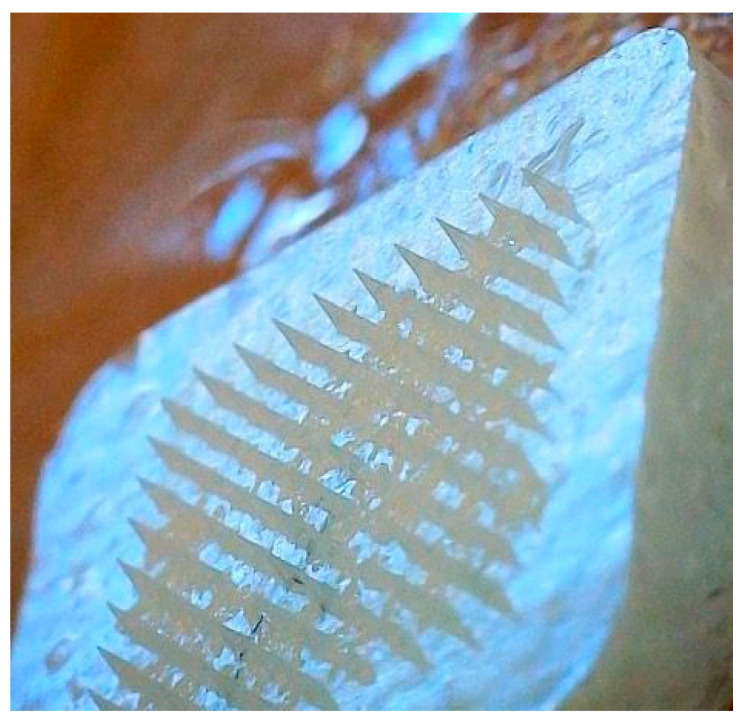
Wi-Fi digital microscopic image of nanoparticles in effervescent microneedles.

**Figure 7 pharmaceutics-16-00308-f007:**
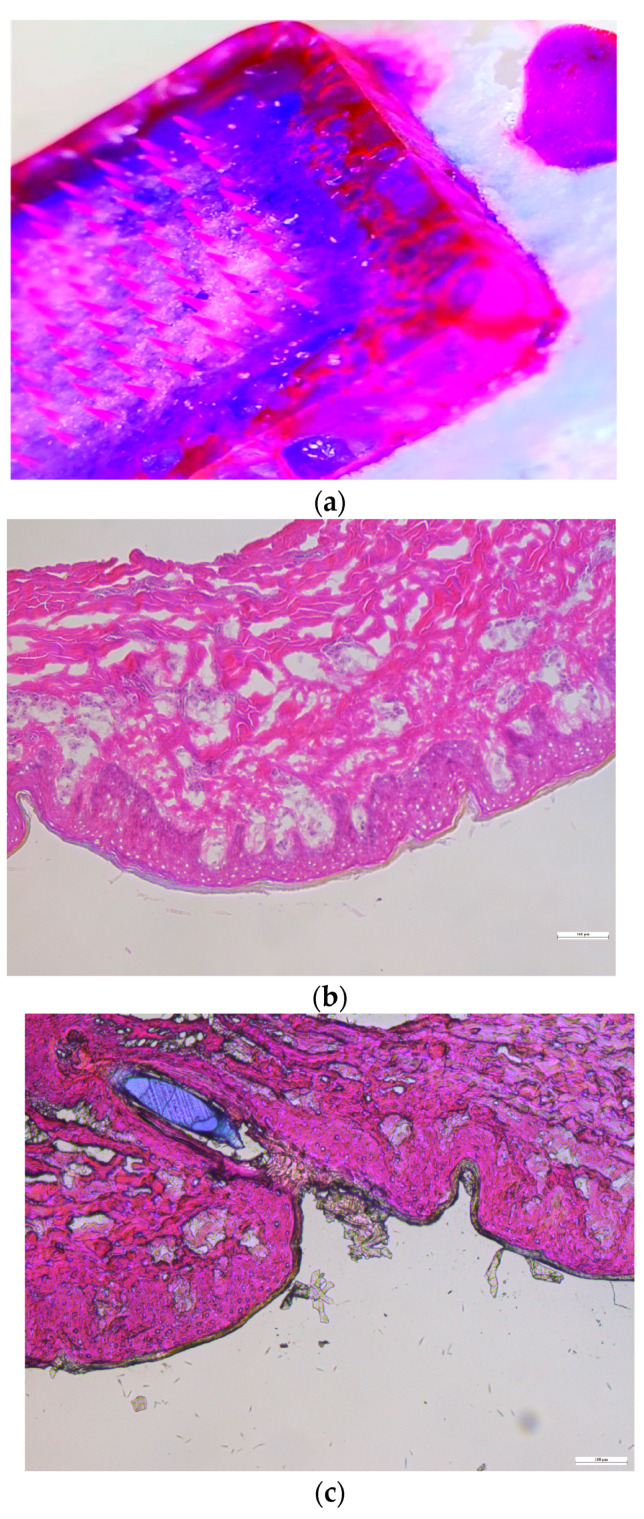
Histological characterization: (**a**) Nile red-incorporating microneedle, (**b**) intact dermatomed porcine ear skin, and (**c**) dermatomed porcine ear skin treated with microneedles.

**Figure 8 pharmaceutics-16-00308-f008:**
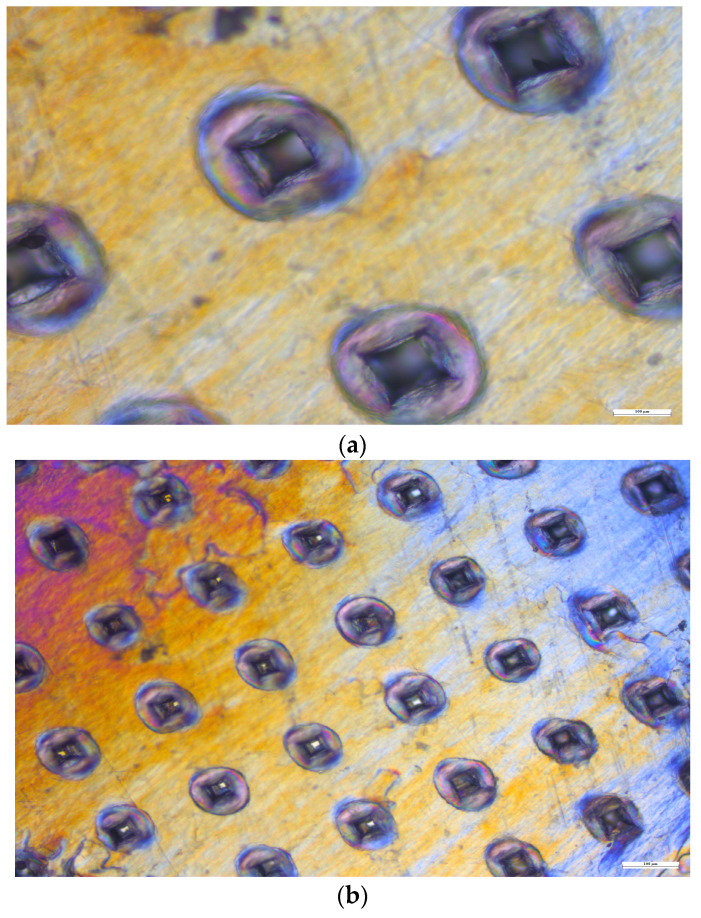
Microscopic views of the Parafilm M^®^ layers. (**a**) First layer under 10× magnification. (**b**) First layer under 4× magnification. (**c**) Second layer under 10× magnification. (**d**) Second layer under 4× magnification. (**e**) Third layer under 4× magnification. (**f**) Fourth layer under 10× magnification.

**Figure 9 pharmaceutics-16-00308-f009:**
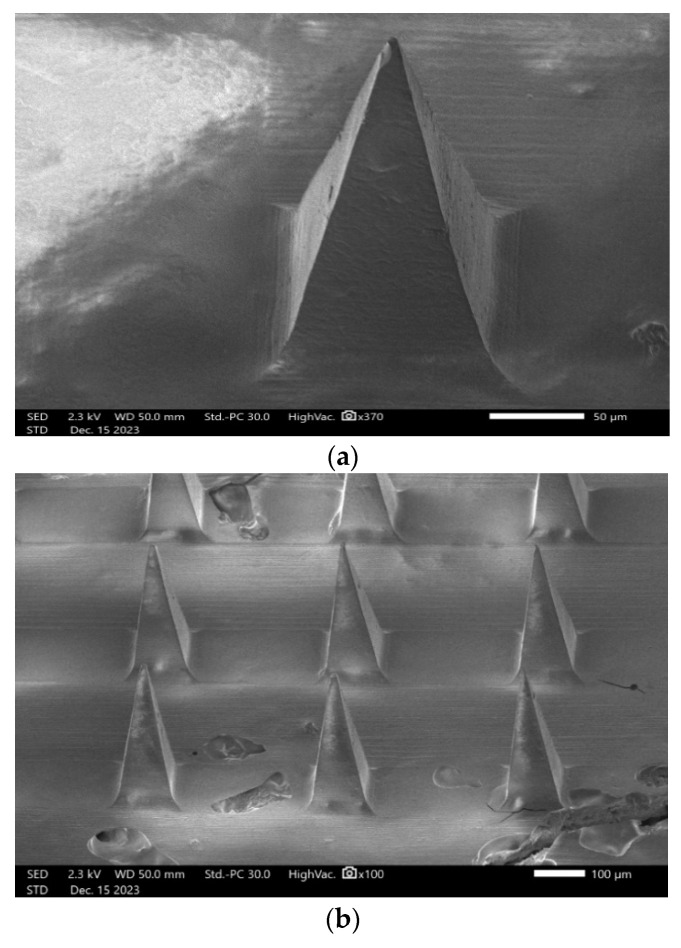
Scanning electron microscopy (SEM) images showing (**a**) single lurasidone nanoparticles in a microneedle, (**b**) lurasidone nanoparticles in microneedles, and (**c**) lurasidone in microneedles.

**Figure 10 pharmaceutics-16-00308-f010:**
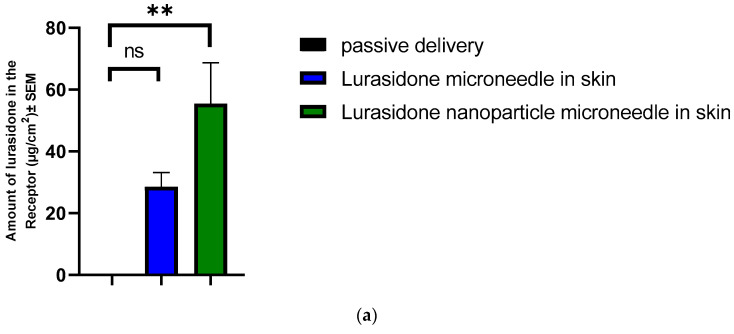
Evaluation of drug delivery from microneedles through the skin to the receptor: (**a**) cumulative amount of lurasidone delivered into the receptor after 3 days; (**b**) amount of lurasidone delivered into the skin at the end of 3 days; and (**c**) permeation profile of lurasidone. Statistical analysis performed with one-way ANOVA; ns indicates no significant difference between groups, * and ** indicate significant difference between groups, *p* < 0.05.

**Table 1 pharmaceutics-16-00308-t001:** Optimization parameter of nanoparticles.

Organic Phase	Aqueous Phase			Organic Phase/Aqueous Phase (PVA 1%)				Drug/Polymer Ratio					
THF	1%	2%	3%	1:15	1:20	1:30	1:35	1:5	2:5	3:5	1:3	2:3	1:1
Acetone	1%	2%	3%	-	-	-	-	-	-	-	-	-	-
DCM	1%	2%	3%	-	-	-	-	-	-	-	-	-	-

**Table 2 pharmaceutics-16-00308-t002:** Various combinations of nanoparticles formulated in 1% *w/v* PVA (NP solution) alongside different concentrations of PVP.

Wt. ratio of NP: PVP	1.5:1	1:1	1:1.5	1:2
NP solution	1.5	1	1	1
20%,15%, and 10% PVP	1	1	1.5	2

**Table 3 pharmaceutics-16-00308-t003:** Solubility of lurasidone in different solvents.

Solvent	Solubility (mg/mL) n = 3
PBS	0
PEG	3975.1 ± 45
propylene glycol	954.55 ± 34
Volpo 6%	98.3 ± 12
80% PEG: 20% Volpo 6%	494.6 ± 15
70% PEG: 30% Volpo 6%	121.7 ± 17
10 mM citric acid buffer pH 3	15,067.4 ± 46
10 mM citric acid buffer pH 3.5	3603.4 ± 24
10 mM citric acid buffer pH 4	1048.2 ± 18
10 mM citric acid buffer pH 4.5	321.3 ± 16

**Table 4 pharmaceutics-16-00308-t004:** Comparison of nanoparticle characteristics by fabrication method and solvent.

Preparation Method	Nanoparticle Size	Polydispersity Index	% of Drug Encapsulation
Acetone and homogenization	325.3 nm	0.295 Mw/Mn	%28.3
Acetone and syringe pump	284.8 nm	0.163 Mw/Mn	%40.2
THF and homogenization	335.8 nm	0.432 Mw/Mn	%31.8
THF and syringe pump	409.2 nm	0.462 Mw/Mn	%57.3

**Table 5 pharmaceutics-16-00308-t005:** Different PLGA to drug ratios for the optimization of the nanoparticle formulation.

PLGA/Drug	Nanoparticle Size	Polydispersity Index	% of Drug Encapsulation
5:1	332.7 nm	0.227 Mw/Mn	%50.1
5:2	367.9 nm	0.159 Mw/Mn	%49.0
5:3	877.0 nm	0.592 Mw/Mn	%36.7
3:1	316.1 nm	0.298 Mw/Mn	%54.3
3:2	635.2 nm	0.383 Mw/Mn	%25.3
1:1	643.6 nm	0.449 Mw/Mn	%42.2

**Table 6 pharmaceutics-16-00308-t006:** Saturation solubility of lurasidone in different polymers.

Solution	µg of Drug Quantity in 75 µL
Just PVA	61.412
5% *w*/*v* PVP	54.916
10% *w*/*v* PVP	35.988
20% *w*/*v* PVP	19.137

## Data Availability

The data supporting this article are found within the text. Any additional data and the data that support the plots within this paper are available from the corresponding author upon reasonable request.

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
