# Peer review of "Microneedle-Assisted Transdermal Delivery of Lurasidone Nanoparticles"

_pharmaceutics, 2024, doi:10.3390/pharmaceutics16030308_

Round 1
Reviewer 1 Report
Comments and Suggestions for Authors
The study explains the formulation of effervescent microneedle assisted delivery of drug laoded nanoparticles. Synthesis of effervescent transdermal patch is obtained, characterized and tested in vitro. The study in general lacks rationale, and does not articulate well as a single subject. Many experiments are performed without any rationale/need and are not discussed well in the study. Moreover, the language at times is hard to understand and requires significant modifications. More specific comments are as follows:
Introduction: this section needs summarization and rewriting, the introduction is not well-articulated and reads as separate paragraphs relevant to different fields of science. This section needs to be narrowed down to the relevant recent research on microneedling based technologies and should not be focused on nanotechnology in general. This is too broad a topic for an intro of an original article.
Moreover, last paragraph should be modified to include broad overview of experiments performed in this study for the evaluation of transdermal patch.
Methods: this section should be reorganized to reflect the actual work flow: synthesis of nanoparticles, drug loading, transdermal patch formation and then bioanalysis of the patch. Methods section must be modified to include specific information for each experiment that can be reproduced by another researcher in the field. For example, what was amount of PLGA (in mg) added in 500 uL solution to make nanpoparticles. What was the amount of drug in mg? What kind of microscopic examination was conducted during the synthesis of PLGA nanoparticles? Where % of certain solution is indicated, explain if this is a weight % of mole % compared to certain chemical? Provide details of DLS and zeta analysis, what was the standard used for equipment calibration? What fitting model for used to obtain size and charge? Model and Name of the equipment? For HPLC studies, what was the solvent and method, type of column, equipment manufacturing details? Section 2.2.10, the need for this experiment is unclear, modify the text to make this obvious as to why this study is needed? Section 2.2.11, what nanoparticles is this study looking at? Section 2.2.12, how was the device designed? Any computer softwares? What was the concentration of nanoparticles applied in section 2.2.12? Same is true for other sections, provide experimental details.
For animal experiments, please include any bio/research ethics approval that was obtained?
Results: Section 3.1 please explain what is being analyzed by HPLC? Why was solubility of drug in different solvents was tested? When drug release would only occur under physiological conditions? Again rational for some of the experiment is missing and this is hard to follow the study. Same suggestions are applied to other results sections.
What is the effect of chemical enhancer on skin damage? Were any viability test performed? Figure 3 and 5-7 scales are missing on microscope images.
Comments on the Quality of English LanguageNeed major changes/rewriting
Author Response
Thank you for the comments, please see the attachment.

Reviewer 2 Report
Comments and Suggestions for Authors
This manuscript deals with the development of an effervescent microneedle patch aimed for the systemic release of lurasidone within three days for the treatment of schizophrenia. Although the idea is interesting and very actual, this manuscript has some important limitations that limit its acceptance in the present form. Point by point comments are listed below:
1. The abstract is too general. More details are needed on the microneedles and nanoparticles prepared, the methodology used for characterization and results obtained.
2. The introduction is also quite general and contains a lot of well-known information. It should focus more on the microneedle effervescent patches, the type of nanoparticle used and the advantages of combining these two technologies (the description of schizophrenia, skin and drug delivery strategies should be shortened). Furthermore, as different approaches have been tested to improve the delivery of lurasidone through the skin (chemical penetration enhancers, solid and effervescent microneedle patches in combination with PLGA nanoparticles), the aim of the study should be restated and expanded to better understand the rationale for the present investigation.
1. Details of the equipment used should be added where they are missing (e.g. type of HPLC, centrifuge, vertical static Franz diffusion cells, zetasizer). Also, please specify which microscope was used during the nanoparticle preparations.
2. Franz diffusion cell experiments: thickness of dermatomed skin and receptor volume have to be added. Please provide data regarding the chemical stability of lurasidone during 72 h in the selected medium.
3. The authors utilized their own procedure for in vitro release testing, analyzing the amount of lurasidone released from 10 mg of nanoparticles added to a vial with 1 mL of citric acid over period of 72 hours. As no membrane was used, it is essential to demonstrate that the tested formulation remained unchanged throughout the testing period.
4. Description of abbreviation NMP solution should be added.
5. The properties of prepared effervescent microneedles (length, density, width, shape) have to be added.
6. 3.2 The section solubility study should be part of the IVPT/IVRT study, as only data on the solubility of lurasodine in different receptor media were presented.
7. 3.4. Delivery with chemical enhancers: The authors has written that the permeation of lurasodine from the solvent systems tested did not reach the desired target, but it is not clear what the target was. Furthermore, it is not recommended to compare the drug permeation from diverse formulations containing different amounts of the tested drug. In other words, the differences observed are entirely to be expected.
8. Figure 2b. Authors showed the amount of lurasidone delivered into the skin at the end of 3 days, but there is no description of the protocol for this test. In addition, the SD (or SE) should be added to points at Figure 2c. 4 repetitions for this type of study is very low (the authors probably presented the SE due to the high SD)
9. 3.7. IVPT with Dr. Pen TM Ultima A6. The authors also used stainless steel microneedles to create microchannels in the skin. The purpose of this study is not entirely clear. The results obtained were not compared with the novel effervescent microneedle patch.
10. No results of in vitro release experiments were shown (the discussion of these results is not supported by the results). Due to the numerous methodological problems with this study, I suggest deleting this test.
11. 3.9. Lurasidone Concentration in Microneedle Solution: What was the target dose? How was it selected? How can one be sure that the amount of drug loaded into microneedles would be completely available into the systemic circulation. The equipment used for homogenization should be added. Furthermore, the microscopic image in Figure 5b clearly shows that lurasodine was not dissolved (it was transformed into a nanocrystalline suspension).
12. 3.12. Histology study. To enhance the visibility of the microneedles under the microscope during the histology study, Nile red was incorporated into the formulation of microneedles, but no description of their preparation was provided.
13. The image of prepared microneedle patch should be presented.
14. 3.16. IVPT study with microneedles: The authors showed only the amount of lurasidone delivered into the receptor and the amount of lurasidone delivered into the skin at the end of 3 days. What has happened in meantime? The permeation profiles need to be presented.
15. The authors concluded that a microneedle loaded with lurasidone nanoparticles was able to reach the therapeutic target for lurasidone. However, it should be kept in mind that this study was performed in vitro, using excised porcine ear skin, without circulation.
Comments on the Quality of English Language
Minor editing of English language required.
Author Response

(The authors gave the same response as above.)

Reviewer 3 Report
Comments and Suggestions for Authors
This manuscript designed a long-acting microneedle patch, used microneedle-assisted transdermal delivery of lurasidone nanoparticles, and conducted relevant in vitro percutaneous experimental studies. It has definite innovation and subsequent clinical application value and provides a detailed description and analysis. However, there are still some contents that need to be improved. There are some suggestions as follows:
1. In the "1. Introduction" section, the 115th line mentions "effervescent microneedle patches", and this type of microneedle was also used in the study of this manuscript. However, the proposal, development, and application of this type of microneedle have yet to be described. I suggest adding some relevant content in this section.
2. In "3.5." In the "Fabric of lurasidone nanoparticles" section, there are multiple selections and determinations of optimal parameters. I suggest adding some tables to summarize and compare the experimental results of each parameter more intuitively.
3. What device captured the "Figure 3, Microscopic view of Lurasidone nanoparticles" image? The quality of the picture shooting is not very high, and there is no ruler, so it needs to be modified.
4. In the "3.8. Volume of microneedle tips" section, lines 527-529, " To further enhance the amount of lurasidone in the solution, 5% transcutol was incorporated. Transcutol was selected based on its proven efficacy in improving lurasidone delivery, as observed in studies comparing various chemical enhancers." Is this a reference or a conclusion drawn from previous experiments? The references need to be annotated, and the previous experiments need to be explained.
5. Figure 5 does not have a ruler.
6. There is no ruler in Figure 6, and it seems that the magnification in Figures 6 b and 6 c is inconsistent.
7. In Figure 7, some figures do not have rulers, and the figures with rulers do not specify the specific dimensions of the rulers.
8. In the "3. Results "section, the difference between effervescent microneedles and other microneedles lacks relevant validation.
9. In the "3. Results " section, there is a lack of relevant validation for the solubility of microneedles.
Author Response

(The authors gave the same response as above.)

Reviewer 4 Report
Comments and Suggestions for Authors
The manuscript describes a detailed study using microneedles to deliver nanoparticles. Overall, the manuscript presents the results obtained clearly, and the results are analyzed and described in a scientific manner. A few improvements could be:
Solubility studies were performed at ambient temperature. Why was the temperature not controlled or even the measured temperature reported?
Figure 5 does not have a scale bar
Comments on the Quality of English LanguageEnglish is fine.
Author Response

(The authors gave the same response as above.)

Round 2
Reviewer 1 Report
Comments and Suggestions for Authors
The authors has made the suggested changes, I suggest publishing the study in Pharmaceutics.
Comments on the Quality of English Languagen/a
Reviewer 2 Report
Comments and Suggestions for Authors
The authors have significantly improved the quality of manuscript. There are only two small remarks:
1. Please check data in Table 5. The values of polydispersity index are strange.
2. The same abbreviations for standard error should be used at all graphs.
